# Single-cell measurement of plasmid copy number and promoter activity

Bin Shao[1], Jayan Rammohan [2], Daniel A. Anderson [1], Nina Alperovich[2], David Ross [2] & Christopher A. Voigt [1✉]

Accurate measurements of promoter activities are crucial for predictably building genetic systems. Here we report a method to simultaneously count plasmid DNA, RNA transcripts, and protein expression in single living bacteria. From these data, the activity of a promoter in units of RNAP/s can be inferred. This work facilitates the reporting of promoters in absolute units, the variability in their activity across a population, and their quantitative toll on cellular resources, all of which provide critical insights for cellular engineering.

[1] Synthetic Biology Center, Department of Biological Engineering, Massachusetts Institute of Technology, Cambridge, MA, USA. [2] Biosystems and Biomaterials Division, Material Measurement Laboratory, National Institute of Standards and Technology, Gaithersburg, MD, USA. ✉email: cavoigt@gmail.com

Accurate measurements of promoter activities are crucial for understanding the biophysics of transcription and enable the predictive construction of genetic systems[1]. The strength of a promoter can be described as the flux of transcribing RNA polymerases (RNAPs) exiting a promoter (RNAP/s)[1]. In essence, this is the time- and population-averaged output of the biophysical complexity of an individual promoter, which involves abortive short transcripts, pausing, and bursts of activity[2–4].

In the design of genetic circuits, promoters carry the signal between sensors and gates[5–7]. In metabolic engineering, enzyme levels can be balanced by selecting promoters of different strength[8]. Increasingly, these tasks are being performed by computer aided design (CAD), the precision of which is limited by part measurement accuracy[7,8]. The strength of promoters in absolute units is rarely known and is more often measured indirectly with a reporter gene and provided in "arbitrary units"[9]. It has been proposed to define a constitutive promoter ($P_{J23101}$) carried on a p15A plasmid as a reference, to which a promoter-of-interest is compared and the strength reported in relative promoter units (RPUs)[10]. The conversion of 1 RPU to RNAP/s has been estimated[7,10].

Single-molecule methods to visualize transcribing RNAPs exiting a promoter are either performed in vitro or are technically challenging[11–14]. Across a population, the same promoter may have different activities in each cell because of extrinsic noise; in other words, differences in cellular resources such as RNAPs and plasmids[15–17]. Promoter activity can be inferred from mRNA transcripts, but methods such as single molecule fluorescence in situ hybridization (smFISH) require fixing the cells and single-cell RNA-seq loses the required resolution[4,11,18–21].

A problem with using bulk measurements to calculate promoter activity is that each cell has a different number of copies of the promoter because of differences in the copy number of the plasmid or genome on which it is carried. Plasmid copy number is dictated by its replication origin and can change depending on the genetics of the host strain[22] and growth conditions[23] and the copy number of the genome varies depending on the distance to the origin and growth rate[24,25]. The average plasmid copy number has been estimated with bulk DNA measurements, but no method has been developed to count the plasmid copy number in single living cells. Plasmids can be visualized using DNA FISH and super-resolution microscopy or by fusing DNA-binding proteins to fluorescent reporters, but they have not been calibrated to provide absolute units[26–33]. PCR-based methods suffer from accuracy and is difficult to implement for single cell measurements[34,35].

In this work, we develop a method to use fluorescent reporters fused to binding proteins that label the plasmid and RNA transcripts so that they, along with protein expression, can be measured simultaneously in individual cells. Inspired by earlier work to use DNA-binding proteins to detect plasmids in vivo[28–33], we use PhlF fused to red fluorescent protein (RFP) to count plasmids. In the same cell, we use the PP7 RNA-binding protein fused to cyan fluorescent protein (CFP), selected because it requires the fewest repeated operators and has minimal impact on mRNA degradation[36–46]. These data enable us to calculate the activity of promoters in absolute units in individual living cells. Further, we are also able to simultaneously detect protein expression using yellow fluorescent protein (YFP).

## Results

### Quantification of the plasmid copy number in single cells.
Plasmids are detected in individual cells using fluorescence microscopy (Fig. 1a, b). The backbone is modified to insert a region with 14 operator repeats that bind the PhlF repressor[47], flanked by strong terminators (Fig. 1a and Supplementary Fig. 2). Note that this is much smaller than previous work; for example, the use of 240 tetO sites to bind TetR[26,28]. This plasmid also contains the green fluorescent protein gene (gfp) under the control of a constitutive promoter, allowing for the simultaneous measurement of plasmid copy number and protein expression. From a second plasmid (pSC101 origin), a PhlF-RFP fusion protein is expressed under the control of an aTc-inducible promoter. The plasmids are transformed into E. coli NEB 10β and grown at 37 °C in M9 media until reaching exponential phase in the presence of aTc, ampicillin and kanamycin (Methods). An aliquot of cells is taken, placed on a cover slip with an agar slab and imaged using an inverted fluorescence microscope.

The copy number was determined for different origins of replication (ori) (pSC101, p15A, pColE1, and pUC). (To measure pSC101, the phlF-rfp expression cassette is on the same plasmid, Supplementary Fig. 2). As the copy number increases, punctate red spots become brighter and more abundant (Fig. 1b, Supplementary Fig. 1 and Supplementary Fig. 2) (Methods). When the spot intensity data are plotted for the lowest copy plasmid (pSC101) as a histogram, equidistant peaks are apparent (Fig. 1c). The mean distances between the first four peaks were used to determine the spot intensity due to one plasmid. Then, the mean plasmid copy number per cell is calculated for each backbone: 4 (pSC101), 9 (p15A), 18 (pColE1), and 61 (pUC) (Fig. 1d and Supplementary Fig. 2). These means are consistent with previous bulk measurements[34,48]. The numbers of plasmids per spot were also consistent with previous measurements of plasmid clusters within cells; for example, for pColE1, each plasmid spot contains 11 plasmids on average, which is similar to the cluster size (~10) revealed by quantitative localization microscopy[18]. Plasmid clustering could be explained by multi-merization, in which plasmids form high order oligomers as a result of recombination between individual plasmid molecules, or the sharing of replication machinery[49].

The distribution of plasmid copy numbers across a population of cells is shown in Fig. 1e. The distributions are wide, with standard deviations on the order of the mean copy numbers (4, 11, 15 and 60 for pSC101, p15A, pColE1, and pUC). The extreme ends of the distributions were notable, with a large fraction of cells lacking plasmid entirely (5%, 3%, 1% and 1%) and many cells containing 4-fold or more of the mean copy number. We found these distributions were consistent between day-to-day measurements, antibiotic choice, and protein expression cassettes (Supplementary Fig. 3). Even the incorporation of a toxin-antitoxin system (hok/sok), which has been used to minimize plasmid loss[50], results in the same distribution.

We developed a mathematical model to determine if these distributions would emerge from simple rules for plasmid replication and cell division (Fig. 1f) (Methods and Supplementary Note 1). Plasmids are first distributed across a population for a desired average copy number $N_0$. Then, iterations of growth and division are performed: cells are selected randomly, plasmids are distributed following a modified binomial distribution (partitioning coefficient, $a$)[51,52], and they replicate with feedback[51,53] (sensitivity, $K$) until $N_0$ is reobtained. The shape of the distribution is determined by the tightness of plasmid copy number control and partitioning, with more even partitioning leading to less plasmid loss and tighter copy number control resulting in lower variance of the distribution. With two fit parameters ($a$ and $K$), this converges on the observed plasmid distributions from disparate starting distributions (Supplementary Fig. 4). The prediction errors could be due to additional sources of randomness in plasmid segregation. The parameter values

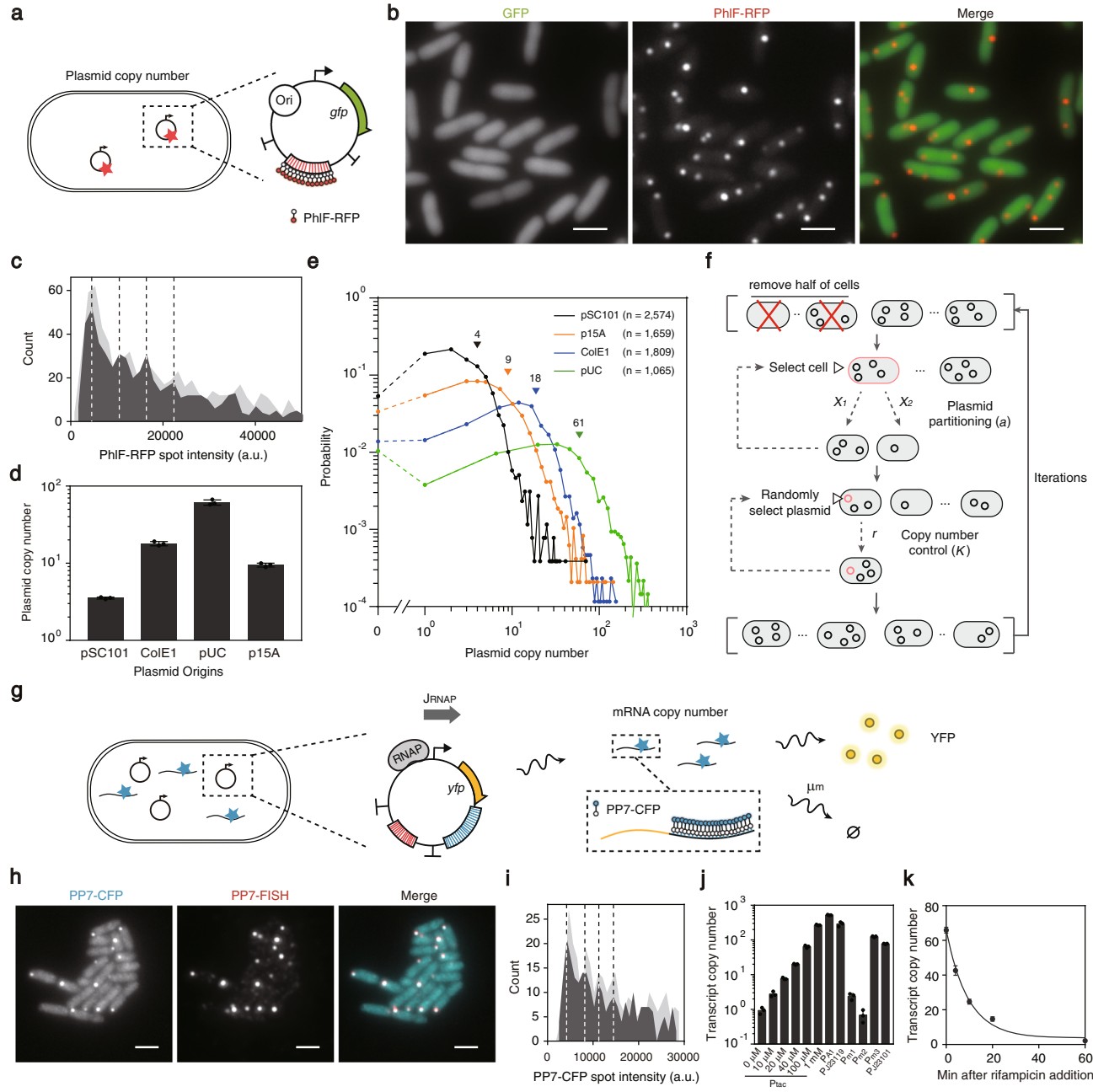

reflect known plasmid replication and segregation mechanisms for different backbones (Supplementary Table 1).

**Quantification of the transcript copy number in single cells.** An approach was adopted to visualize the number of mRNA transcripts in a cell that is compatible with the plasmid copy number measurement. A second repeat region was placed in the target plasmid containing 20 copies of a PP7 stem loop[37] (Fig. 1g). This is inserted after *yfp* and before the terminator so that the mRNAs contain the loops. From the second plasmid, a PP7 fusion to cyan fluorescent protein (CFP) is expressed under the control of an aTc-inducible promoter (Supplementary Fig. 5). A strong ribosome binding site (RBS) was selected as we found that higher PP7-CFP expression was required to quantify larger transcript numbers, and these high levels were not detrimental to quantifying low transcript numbers (Supplementary Fig. 5). The transcripts were visualized for a set of constitutive and inducible promoters carried on the p15A target plasmid (Supplementary

Fig. 6). Very bright foci were observed for strong promoters and transcripts can be barely visualized for weak promoters. To validate that the spot signal corresponds to labeled transcripts, we performed FISH experiments using probes hybridizing to the PP7 binding sites (Fig. 1h) (Methods).

The PP7-CFP spot intensities were then used to calculate the absolute number of transcripts per cell. Following an earlier approach[40], we constructed a histogram of spot intensities, where each peak corresponds to an additional transcript in the cell (Fig. 1i and Supplementary Fig. S6). From this, the intensity corresponding to a single transcript can be calculated and used to calculate the absolute number of transcripts in each spot, which is further summed for each cell to obtain the total transcript number per cell (Fig. 1j). The transcript copy number distributions are wide, with standard deviations greater than the mean copy number (Supplementary Fig. 6). Similarly broad distributions have been reported for IPTG-inducible promoters using smFISH[4,54] and bimodality has been observed for other *E. coli*

**Fig. 1 Measurement of plasmid copy number and transcript number. a** Schematic of the plasmid copy number calibration construct. PhlF-RFP is expressed from a second plasmid (pSB235-237). "Ori" indicates where the plasmid origin is changed in the plasmid being measured. Plasmid maps and part sequences are provided in Supplementary Fig. S18 and Table S3. **b** Images are shown for the measurement of ColE1 plasmids (pSB220) in the cell. GFP is the expression of the reporter gene (525/50 nm) and PhlF-RFP shows the binding of the reporter to the plasmids, where brighter spots indicate more plasmids in a cluster (645/75 nm). Scale bar, 2 μm. **c** The histogram of PhlF-RFP spot intensity from the low copy pSC101 backbone is shown. The dashed lines show the peaks, the distance between them is used to calculate the spot intensity of individual plasmids. The light and dark gray histograms show the distributions from replicates performed on different days. **d** Measured mean copy number for different plasmid backbones. The means are calculated from three replicates performed on different days and the error bars are the standard deviations of these experiments. **e** Copy number distributions across a population of cells are shown. The triangles indicate the means of the population. The distributions are made from a combination of 3 replicates performed on different days (cells numbers from each replicate is 858 for pSC101, 553 for p15A, 603 for ColE1 and 355 for pUC, p values from the two-sample Kolmogorov–Smirnov test for pooling the replicates are 0.97/0.58/0.10 for pSC101, 0.89/0.97/0.67 for p15A, 0.76/0.22/0.72 for ColE1, 0.61/0.98/0.33 for pUC. **f** The model for simulating the convergence onto a plasmid copy number distribution (Supplementary Note 1). The two daughter cells get $x_1$ and $x_2$ plasmids. $a$ is the partitioning coefficient and $K$ is the sensitivity for plasmid replication control. **g** Schematic showing the detection of mRNA in single cells. PP7 binding sites (blue) are shown on the pSB223 plasmid and mRNA. This plasmid is co-transformed with pSB233, which contains the genes for PP7-CFP and PhlF-RFP expression (not shown). **h** Overlap of the FISH signal targeting PP7 binding sites with the PP7-CFP signal (Methods). Scale bar, 2 μm. **i** The histogram of PP7-CFP spot intensity from a plasmid expressing small amounts of mRNA is shown ($P_{tac}$/20 μM IPTG/pSB208). The dashed lines show the peaks, the distance between them is used to calculate the spot intensity of individual mRNAs. The light and dark gray histograms show the distributions from replicates performed on different days. **j** Quantification of mRNA copy number for a library of promoters on p15A plasmid backbone. Plasmid maps and part sequences are provided in Supplementary Fig. S18 and Table S3. **k** Dynamics of transcript copy number from the constitutive promoter $P_{J23101}$ after rifampicin addition. The data was fitted to a single exponential decay. The resulting degradation half-life (μ) is 6.8 ± 0.3 min. The impact of rifampicin addition on YFP expression and plasmid copy number is shown in Supplementary Fig. S10. For parts b and h, microscope experiments were repeated three times with similar results. For parts j and k, the means were calculated from the population means from three replicates performed on different days and the error bars represent the standard deviation of these means. Source data are provided as a Source Data file.

promoters[55,56]. No crosstalk is observed between the DNA- and RNA-binding fusion proteins (Supplementary Fig. 7); therefore, they can both be used to simultaneously measure DNA and mRNA copy numbers in single cells. In addition, this technique can be used to label other RNAs; for example, the small guide RNA (sgRNA) that bind to dCas9 (Supplementary Fig. 8).

Our calculation of the fractions of cells containing no mRNAs or plasmids could be due to less-than-perfect detection efficiency. We sought to determine whether this could be due to cells that have the target molecule, but did not get labeled due to a fluctuation in the labeling protein. Cells for which we found no mRNA transcripts do not have a lower total PP7-CFP expression (Supplementary Fig. 6e). Changing the PP7-CFP expression level does not change the fraction without mRNA: 5% when high and 6% when low (Supplementary Fig. 6f). Further, these numbers are consistent with previous results from smFISH experiments (5–8%)[11,55]. The FISH intensity and PP7-CFP spot intensity are strongly correlated ($R^2 = 0.95$ Supplementary Fig. 6). Similarly, the number of plasmids per cell and the fraction of cells without plasmid is not impacted by the total PhlF-RFP expression (Supplementary Fig. 2j, k). Cells without detectable PhlF-RFP spots also have very low GFP expression, supporting that it is the result of plasmid loss rather than detection error (Supplementary Fig. 2).

**Single cell measurement of promoter activity**. Inferring the activity of a promoter in units of RNAP flux requires promoter copy number, mRNA copy number and degradation rate. The productive RNAP flux is the same as the generation rate for complete mRNAs. The flux in units of RNAP per second per DNA (RNAP/s-DNA) is calculated as $\bar{J}_{RNAP} = \tau(m/N)$, where τ is the mRNA degradation rate, $m$ is the mRNA copy number and $N$ is the plasmid copy number. The half-life of mRNA ($\mu = \ln(2)/\tau$) in *E. coli* is typically in the range of several minutes[22,23]. Using a rifampicin assay (Methods), we measured a mean half-life of 6.8 ± 0.3 min for transcripts from $P_{J23101}$ driving *yfp* and the PP7 repeat in the presence of PP7-CFP (Fig. 1k, Supplementary Fig. 9, Supplementary Fig. 10). Note that using a population-averaged mRNA degradation rate is an approximation because of extrinsic

noise in degradation machinery and variability in mRNA partitioning[41,57].

From these data, we calculated $\bar{J}_{RNAP}$ from the reference promoter $P_{J23101}$ in individual cells (Fig. 2a and Supplementary Fig. 11). The fluorescence distribution from this promoter (YFP) and cell growth rate are not affected by the addition of the binding operators nor co-transforming with the plasmid containing the fusion proteins. The promoter activity is measured in each cell by simultaneously measuring the mRNA and plasmid copy numbers (Fig. 2b), while assuming the degradation rate is constant. The distribution of promoter activities for a population of cells in the same experiment is shown in Fig. 2c (only for those cells in which DNA is detected). The population mean is $<\bar{J}_{RNAP}> = \tau<m/N> = 0.019$ RNAP/s – DNA, similar to the values estimated in the literature[7,10]. However, the distribution is broad: the standard deviation across a cell population from a single experiment is 0.028 RNAP/s-DNA whereas the standard deviation of the population means from measurements from different days is 0.002 RNAP/s-DNA (Fig. 2c). The population is bimodal because no mRNA is detected in 5% of the cells in which plasmids can be detected. This bimodality is not observed in protein expression (YFP) either by microscopy or cytometry (Fig. 2d). This could be explained by the long half-lives of reporter proteins, which averages out the fluctuations in mRNA numbers. Our results demonstrate that the measurement of a highly-expressed fluorescent reporter does not capture the underlying population behavior.

We took time lapse movies of bacteria growing on an agar pad (Supplementary Fig. 12). For each cell, the dynamics of RFP, GFP, and YFP were measured, setting cell division to $t = 0$. The plasmid and mRNA copy number decrease after cell division and increase during the cell cycle, as expected. The promoter activity is initially 3-fold higher, before converging to the average distribution after 5 min. This observation may be due to divergence from the pseudo-steady-state approximation, where the mRNA and protein levels need to adjust to the new DNA concentration after division.

The promoter activity reports the RNAP flux from a single copy of the promoter. Each plasmid carries one promoter and the sum of their activities in a cell is referred to as the "total promoter activity". Using these definitions, we explored how the promoter

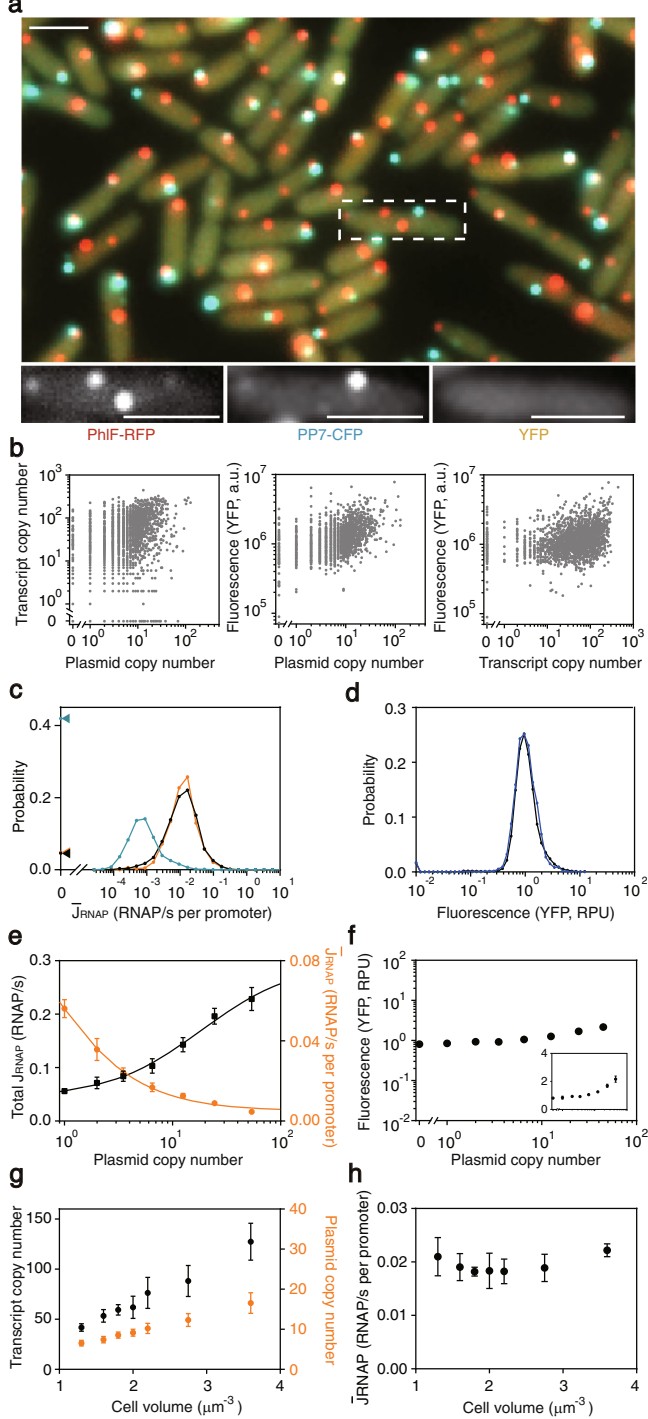

**Fig. 2 Inference of promoter activity in individual cells. a** Sample images showing simultaneous quantification of DNA (RFP), RNA copy number (CFP) and protein production (YFP) for $P_{J23101}$ carried on pSB223 (Supplementary Fig. S1). PP7-CFP and PhlF-RFP are induced with 6 ng/ml aTc (pSB233) (Supplementary Fig. S11). Scale bar, 2 μm. Contrast/brightness for different color channels are changed individually to adjust the color balance (Supplementary Fig. S14). Microscope experiments were repeated three times with similar results. **b** Each point shows a measurement from an individual cell (data obtained from 5 images taken from independent experiments, $n = 2178$ cells). From left-to-right, the $R^2$ to a linear regression model in log scale are: DNA vs mRNA (0.13), DNA vs protein (0.20) and mRNA vs protein (0.14). **c** The calculated promoter activites of $P_{J23101}$ in single cells. The black and blue distributions are when cells are grown in M9 media to exponential phase and stationary phase, respectively (Methods, $n = 2178$ cells and $n = 1863$ cells). The orange distribution ($n = 1344$ cells) is when cells are grown in 2×YT media. The distributions are made from a combination of three replicates performed on different days. Dots are experimental data with lines to guide the eye. Only those cells for which plasmid can be detected are included in the distributions. The triangles indicate the percent of cells where no promoter activity is detected, but where plasmid can be observed. **d** Single-cell YFP fluoresence from $P_{J23101}$ measured by quantitative microscope (black dots) and flow cytometry (blue dots). The medians are scaled so that they are at RPU = 1. The distributions are made from a combination of 3 replicates performed on different days. **e** The total RNAP flux from all promoter copies (black) and the per promoter flux (orange) are shown as a function of the plasmid copy number in individual cells. **f** The YFP expression is shown as a function of the plasmid copy number in individual cells (inset, YFP expression in linear scale). **g** The transcript copy number (black) and plasmid copy number (orange) are shown as a function of the cell volume. **h** The RNAP flux per promoter is shown as a function of the cell volume. For part **e** and **f**, the single cell data is binned by plasmid copy number. The lines show the best fit to a Hill equation. For part g and h, the single cell data is binned by cell volume. For part **e**, **f**, **g** and **h**, data are presented as mean values and the error bars represent the standard deviation of these measurements from three experiments performed on different days. Source data are provided as a Source Data file.

enzyme balancing as large cell-to-cell fluctuations are buffered that would otherwise cause errors. In addition, it speaks to the futility of using high copy number plasmids to increase heterologous protein expression, where there is a diminishing return as well as a disproportionate drain on cellular resources to carry the additional plasmid copies. We also found the per promoter RNAP flux is largely constant for cells of different volumes (Fig. 2h), even when the transcript copy number is highly correlated with the cell volume (Fig. 2g).

Media and growth phase can impact the plasmid copy number. The plasmid copy number is higher (increases to 13) for cells growing in rich media (2×YT) (Supplementary Fig. 14). There's also an increase in the transcript copy number and the promoter activity distribution is similar to the cells growing in M9 media (Fig. 2c). In stationary phase, the promoter activity decreases to $<\bar{J}_{RNAP}> = 0.001$ RNAP/s-DNA and the fraction showing no activity increases to 41%, consistent with σ70 being unavailable (plasmid copy number also increases to 17) (Fig. 2c and Supplementary Fig. 14). As the result of growth arrest, the YFP fluorescence is higher in stationary phase even when transcriptional activity is low (Supplementary Fig. 14).

The system was then used to measure plasmid copy number and the activity of the $P_{J23101}$ in other strains (Fig. 3, Supplementary Fig. 15). First, we tested *E. coli* MG1655, which is closer to wild-type than *E. coli* NEB10-beta. The copy number distributions for the

activity changes as a function of the number of promoters carried in a cell. As expected, the total activity is higher when a cell contains more plasmids (Fig. 2e and Supplementary Fig. 13). For cells with a fixed number of plasmids, the transcript distribution is wider than a Poisson distribution, which could be explained by a two state promoter model[58] (Supplementary Fig. 13). The total promoter activity does not grow linearly with copy number. Rather, the activities of individual promoters decline, which could be due to a limit in the number of available RNAPs in the cell[59,60]. Therefore, a 100-fold change in plasmid copy number is reduced to a 4-fold change in total promoter activity and further reduced to 2-fold in terms of YFP expression (Fig. 2f). This has a profound impact on the use of promoters for genetic circuits or

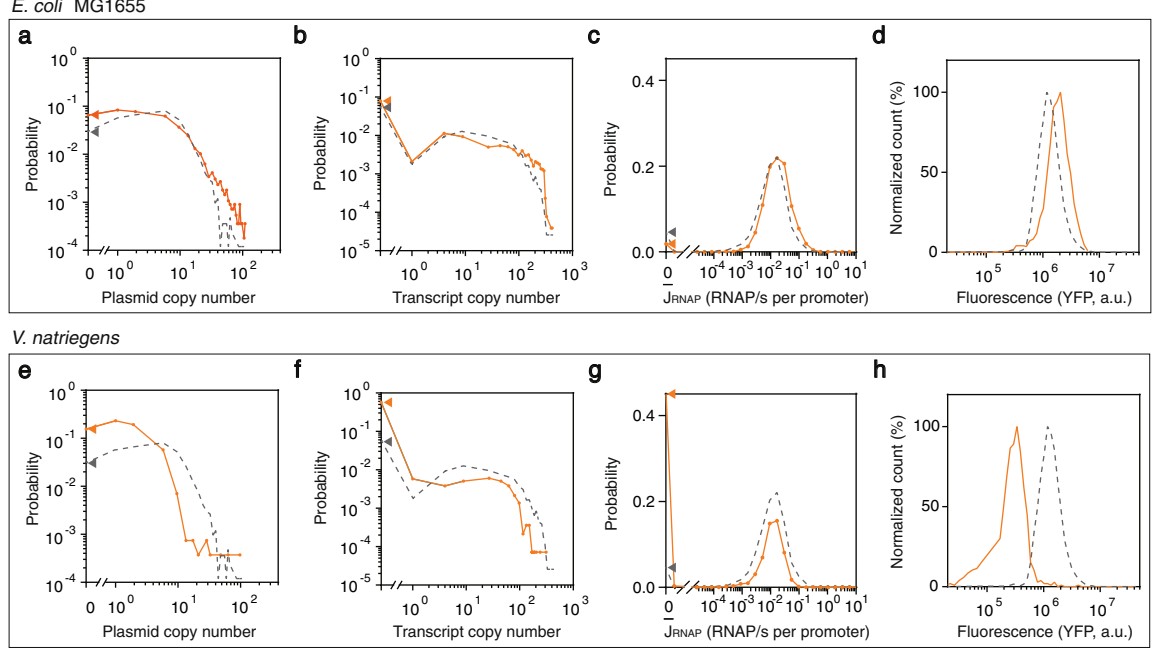

**Fig. 3 Calculation of plasmid copy number and promoter activity across different strains.** The dashed lines in all graphs are the distributions for *E. coli* NEB 10-beta. **a** Plasmid copy number ($n = 1449$, $p$-values from the two-sample Kolmogorov–Smirnov test for pooling the replicates are 0.26/0.92/0.43). **b** mRNA copy number ($p = 0.38/0.99/0.69$). **c** Promoter activites of $P_{J23101}$. **d** The YFP fluorescence distribution. **e** Plasmid distribution for *V. natriegens* ($n = 783$, $p = 0.55/0.05/0.77$). **f** mRNA copy number ($p = 0.25/0.35/0.62$). **g** Promoter activites of $P_{J23101}$. **h** The YFP fluorescence distribution. All the distributions are made from a combination of three replicates performed on different days. For part **a**, **b**, **c**, **e**, **f** and **g**, dots are experimental data with lines to guide the eye. For part **a**, **b**, **e** and **f**, the triangles indicate the percent of cells where no plasmid or transcript is detected. For part **c** and **g**, the triangles indicate the percent of cells where no promoter activity is detected, but where plasmid can be observed. Source data are provided as a Source Data file.

p15a plasmid are very similar (average of $13.2 \pm 1.8$), although plasmid loss is ~2-fold higher, possibly due to the presence of DNA modifying enzymes. The average promoter activity is slightly higher $\langle \bar{J}_{RNAP} \rangle = 0.029$ RNAP/s − DNA. The experiments were then repeated using the fast-growing marine Gram negative species *Vibrio natriegenes*[61]. The p15a plasmid copy number is much lower ($3.0 \pm 0.5$), presumably due to the fast growth rate and the plasmid loss is 8-fold higher (Fig. 3e). The mRNA distribution is slightly lower, but the mean YFP is 4-fold lower. Surprisingly, when the promoter activity is calculated $\langle \bar{J}_{RNAP} \rangle = 0.015$ RNAP/s − DNA, the distribution is nearly identical as the *E. coli* strains. Note that if this promoter were only characterized using the fluorescent output (Fig. 3h), it would have been improperly assigned a lower activity. It is surprising that this reference promoter has similar activities across these strains.

## Discussion

In a single experiment, our approach is able to visualize the number of plasmids, mRNA, and protein expression levels simultaneously in living cells. Using these data, we can measure the promoter activity across a population of cells. There are several caveats including that plasmid and reporter expression that can influence the system, the assumption that mRNA degradation across cells is constant, and the pseudosteady-state approximation that mRNA/protein levels instantly adjust to changes in plasmid copy number and promoter activity.

This reference promoter can be used to convert measurements of other genetic parts (fluorescence or RNA-seq) to absolute units, facilitating their use in biophysical models of system performance[62]. Further, assigning a value to a promoter activity that would otherwise be reported in AU or RPU provides insight into the cellular resources needed to run a system[63,64]. For example, we can now estimate the RNAP usage of a 4-input genetic circuit

used to make an *E. coli* digital display (the circuit referred to as "Segment A") requires a continuous input of 90 RNAP to maintain the lowest off state and 130 RNAP to maintain the highest on state (Supplementary Figure 17)[13,65]. This draws resources from a finite cellular pool of ~5000 RNAPs[13,59]. Having the underlying distribution of promoter activities, rather than just the average, can be used to calculate the failure probability of a system, whether it be for a subset of cells to perform the wrong computation or accumulate a toxic intermediate due to a mismatch in enzyme levels.

## Methods

**Strains and media**. *E. coli* strain NEB 10-beta [Δ(ara-leu) 7697 araD139 fhuA ΔlacX74 galK16 galE15 e14- φ80dlacZΔM15 recA1 relA1 endA1 nupG rpsL (Str^R) rph spoT1 Δ(mrr-hsdRMS-mcrBC)] was used for all cloning and experiments (New England Biolabs, C3019). Note that this strain is deficient for non-specific nuclease and recombinase. *E. coli* strain NEB 10-beta, *E. coli* MG1655 (NCBI U00096.3) and *V. natriegens* (ATCC 14048) were used for DNA/mRNA quantifications. *E. coli* cells were grown in LB Miller broth (Miller, BD Difco, 244620); M9 medium containing M9 salt (Sigma-Aldrich, M6030) and 0.4 % D-glucose (Fisher Chemical, D16-1), 0.2 % casamino acids (BD Bacto, 223050), 2 mM MgSO$_4$ (Affymetrix, 18651), 0.1 mM CaCl$_2$ (Sigma-Aldrich, C1016) and 0.34 g/L thiamine hydrochloride (Sigma-Aldrich, T4625); or 2×YT medium (BD, 244020). *V. natriegens* cells were grown in LB and M9 media with 2% NaCl (VWR, 7647-14-5). 50 μg/ml kanamycin (Gold Biotechnology, K-120), 100 μg/ml ampicillin (Gold Biotechnology, MO, A-301-5) and 25 μg/ml chloramphenicol (VWR, AAB20841-14) were used to maintain plasmids in *E. coli*. 100 μg/ml kanamycin and 10 μg/ml ampicillin was used to maintain plasmid in *V. natriegens*. Isopropyl β-D-1-thiogalactopyranoside (IPTG; Sigma-Aldrich, I6758), anhydrotetracycline hydrochloride (aTc; Sigma-Aldrich, 37919), vanillic acid (Van; Sigma-Aldrich, 94770) and 3-oxohexanoyl-homoserine lactone (3OC6-AHL; Sigma-Aldrich, K3007) were used to induce gene expression. 500 μg/ml rifampicin (Sigma-Aldrich, R3501) was used to inhibit RNA synthesis. 20XPP7 binding site repeats were derived from pCR4-24XPP7SL (a gift from Robert Singer, Addgene plasmid #31864)[37]. Hok/sok toxin-antitoxin pair was PCRed from pSC03[50] (gift from Tal Danino).

**Cell growth**. *E. coli* NEB 10-beta cells containing the plasmids of interest were streaked on LB plates (1.5% Agar; BD, Franklin Lakes, NJ) and grown overnight at

37 °C. Single colonies were inoculated into 150 μl LB in V-bottom 96-well plate (Nunc, 249952) with antibiotics. The plates were sealed with AeraSeal film (Excel Scientific, B-100) and incubated at 1000 r.p.m. and 37 °C in an ELMI shaker (ELMI, DTS-4) overnight. Then the overnight cultures were diluted 178-fold into 200 μl M9 medium in a V-bottom 96-well plate and grown at 37 °C at 1000 r.p.m. in an ELMI shaker for three hours. Then the cells were diluted 667-fold by adding 15 μl of culture to 185 μl M9 media, and then 20 μl of that dilution to 980 μl M9 medium with antibiotics and inducers in deep 96-well plate (USA Scientific, 1896-2000) and grown at 37 °C at 900 r.p.m. in an INFORS-HT shaker (INFORS-HT, Multitron Pro) for 5 h ($OD_{600}$ ~ 0.1) before performing microscopy. To quantify the transcript and plasmid copy number in stationary phase, the overnight culture was diluted 200-fold into 1 ml growth media with appropriate antibiotics in a deep 96-well plate. Inducers were added to the cell culture after 4 h growth at 37 °C. Then the cells were grown for another 5 h to reach stationary growth phase. To quantify the transcript and plasmid copy number in 2×YT media, the overnight culture was diluted 178-fold into 200 μl 2×YT media in a V-bottom 96-well plate and grown at 37 °C for 3 h. The cell cultures were diluted 5,336-fold into 1 ml 2×YT media in deep 96-well plate and grown for 5 h before microscope experiment. The same protocol was used to grow E. coli MG1655. For V. natriegens, the cells were grown overnight in LB media with 2% NaCl. Then, the cell cultures were diluted 100-fold into 200 μl M9 media with 2% NaCl in a V-bottom 96-well plate and grown at 37 °C for 1 h. Finally, the cell cultures were diluted 400-fold into 1 ml M9 medium with 2% NaCl in deep 96-well plate and grown for 3 h before performing microscopy.

**Microscopy assay**. Agarose pads were prepared using M9 medium with 1% agarose (SeaKem, 50004)[66]. The agarose pad was cooled to room temperature before sample preparation. To concentrate cells, 1 ml cell culture was centrifuged at 6000 rcf for 2 min in an Eppendorf microcentrifuge (Eppendorf, 5424). The supernatant was removed and the cell pellet was resuspended in 10 μl M9 medium. A 1.5 μl aliquot was pipetted onto a 22×50 mm cover glass (VWR, 48393-059) and covered by an agarose pad to press the cells onto the imaging surface. Another 22×22 mm cover glass (VWR, 48366-067) was placed on top of the agarose pads to reduce vaporization. The cell sample was put on ice when transporting from cell culture to imaging. Microcopy experiments were performed with an inverted epifluorescence microscope (Nikon Ti-E) equipped with an oil-immersion phase-contrast 100× objective (1.3 NA, CFI Plan Apochromat, Ph3). Images were taken using a fluorescence microscope camera (Andor, DR-328G-CO2-SIL). Nikon Elements software version 4.0 is used to control the microscope and export the images. Four channels are collected, from longest wavelength to shortest wavelength, to minimize crosstalk between different color channels. For the new measurement standard, the signal from RFP was imaged using a 570/40 nm excitation filter, 600 nm beam splitter and 645/75 nm emission filter. The signal from YFP was imaged using a 500/20 nm excitation filter, a 515 nm beam splitter and a 535/30 nm emission filter. CFP-labeled mRNA was imaged using a 436/20 nm excitation filter, a 455 nm beam splitter and a 480/40 nm emission filter. The phase contrast images were acquired using a halogen lamp set to 4 V. For plasmid calibration experiments, the signal from GFP channel was imaged using a 470/40 nm excitation filter, a 495 nm beam splitter and a 525/50 nm emission filter.

**Microscopy Image analysis**. All the images were processed using MATLAB (The Mathworks). Schnitzcells[67] was used to generate cell segmentations from images of color channels that were not used for DNA and mRNA quantification. Images from different color channels were aligned by maximizing their 2-D correlations calculated by corr2 function (MATLAB), which helps to adjust the cell masks for RFP (DNA) or CFP (mRNA) channel. Spot intensities for RFP or CFP channel were quantified using a customized MATLAB script (https://github.com/VoigtLab/Promoter_Activity_Quantification)[66]. A Gaussian filter with a radius of 5 pixels was applied to smoothen the fluorescence profile for each cell (MATLAB function imfilter). Local maxima corresponding to spots were identified using MATLAB function imregionalmax. The pixel values near the maxima were fitted by 2D Gaussian functions with a constant fluorescence background. The fitting was done using MATLAB function lsqcurvefit with default settings. The spot intensities are quantified as the integration of fitted Gaussian functions without the constant background (Figure S1). The total protein expression (fluorescence) is calculated as the sum of all the pixels in the cell. Cell volume is calculated as the cell area multiplies the cell width. The FISH signal was calculated as the sum of all the pixels in the cell.

**Time-lapse measurement**. Overnight cultures of strains were grown in the same manner as for the one-time microscope assay (above). The overnight culture was diluted 1:178 into 200 μl M9 media in a V-bottom 96-well plate, sealed with an AeraSeal film and grown at 37 °C at 1000 r.p.m. in an ELMI shaker for three hours. Then the cultures were diluted 1:400 into 1 ml M9 medium with antibiotics and 6 ng/ml aTc in deep 96- well plate and grown at 37 °C at 900 r.p.m. in an INFORS-HT shaker for 4 h. The agarose pad was prepared following the protocol of Tanenbaum and co-workers[66]. Time-lapse experiments were performed at 37 °C with an inverted epifluorescence microscope (Nikon Ti-E) surrounded by a temperature-controlled enclosure. The enclosure and the imaging platform were

preheated before experiments. The images were taken every 10 min and Nikon PerfectFocus system was used to correct focal drift. Images were acquired and the individual cells between different time frames were manually tracked.

**Statistical analysis**. The following protocol was followed to select the cells from images for analysis. All cells are first identified in the image using Schnitzcells and then a subset of n cells are selected randomly using randperm function (MATLAB) which ensures sampling without replacement. Replicate experiments are performed and the same number of cells are selected from the images obtained as part of each replicate and these are used to create a distribution. The population mean and standard deviation are obtained from this combined distribuiton. For day-to-day variation, the distribution for each replicate is built, the mean calculated and the means for each replicate are used to calculate the standard deviation. To determine whether the replicates are representative of the same underlying distribution, the Kolmogorov-Smirnov Smirnov test is performed for each pair of replicates and the p values are calculated using the kstest2 function (MATLAB).

**mRNA degradation assay**. Overnight cultures of strains were grown in the same manner as for the microscope assay. The overnight culture was diluted 1:178 into 200 μl M9 medium in a V-bottom 96-well plate, sealed with an AeraSeal film and grown at 37 °C at 1000 r.p.m. in an ELMI shaker for three hours. Then the cultures were diluted 1:667 into 1 ml M9 medium with antibiotics and 6 ng/ml aTc in deep 96- well plate and grown at 37 °C at 900 r.p.m. in an INFORS-HT shaker. After 5 h, rifampicin was added to the cultures to a final concentration of 500 μg/ml. The cell cultures were kept in a 37 °C dry bath. At different time points (0 min, 4 min, 10 min, 20 min, 60 min), cells were fixed by adding 500 μl formaldehyde stock (3.5%) to 1 ml cell culture. The cell cultures were vortexed (Scientific Industrial, SI-0236) and placed on ice. Then the cells were washed with cold phosphate buffered saline (PBS; Omnipur, 6505-OP) three times before microscope assay.

**qPCR measurement**. The strains were grown in the same manner as for the microscope assay. After 5-hour growth, 20 μl of cell culture was boiled at 95 °C for 5 min. 1 μl cell lysate was used in a 20 μl reaction system using FastStart Essential DNA Green Master (Roche, 0640271200). Primers that amplify a region of Amp resistance gene and dxs gene in the terminus region of genome was used to quantify copy number of plasmid relative to the copy number of terminus region, which is assumed to be 1 copy per cell[48]. The qPCR experiment was run in a LightCycler 96 with SW 1.1 (Roche) and the plasmid copy number was calculated using the ΔΔCt, assuming an efficiency of 100%.

**Flow cytometry assay**. The cell culture was diluted 1:10 by adding 20 μl of cell culture into 180 μl of PBS containing 2 mg/ml Kan. Fluorescence was measured using the LSRII Fortessa flow cytometer (BD Biosciences). The experiment was run in standard mode at a flow rate of 0.5 μl/s. The FlowJo software version 7.6 (TreeStar) was used to gate the events using forward and side scatter (Supplementary Fig. 16). For each sample, at least 50,000 cells were used for analysis and the median fluorescence value was recorded.

**FISH sample preparation**. Six 20 base-pair oligonucleotide probes that target the PP7 binding sequence were designed using Stellaris Probe Designer version 4.1 (https://www.biosearchtech.com/products/rna-fish/custom-stellaris-probe-sets) and ordered from Biosearch Technologies (Hoddesdon, UK). E. coli NEB 10-beta cells containing the plasmids of interest were streaked on LB plates and grown overnight at 37 °C. Single colonies were inoculated into 4 ml of 2×YT medium in 15 ml culture tubes (Falcon, 352059). The cells were grown overnight for 16 h at 37 °C in an incubator shaking at 300 r.p.m. (Benchmark Scientific, Incu-shaker Mini). Then, the cultures were diluted 1:300 into 25 ml M9 media with antibiotics and inducers in a 50 ml conical centrifuge tube (Falcon, 352070) with a screw-top lid 1/4 closed. The cultures were incubated at 37 °C for 4 h shaking at 300 r.p.m. (Benchmark Scientific, Incu-shaker Mini), after which 4 ml of each culture was aliquoted into each of 5 different 15 ml culture tubes kept in a 37 °C dry bath in a chemical fume hood for measuring 5 different time points. 20 μl of 100 mg/ml rifampicin was added to every 4 ml of cell culture and vortexed for 3 s to mix (Scientific Industries, Vortex Genie). At different time points (0, 2, 4, 20, and 120 min), cells were fixed by adding 2 ml formaldehyde (3.7% by weight, diluted from stock 1:10 into ice cold 1× PBS) to 4 ml cell culture and pipette-mixed. The culture tube was immediately put on ice. After cultures for all time points were fixed and on ice, cells were washed twice with 1×PBS, resuspended in 85% methanol for permeabilization for 1 h at room temperature, and then stored at 4 °C for 2 days (at the permeabilization stage, cells can be stored at 4 °C for up to 1 week). Cells were transferred to new Eppendorf tubes, washed in a solution of 50% formamide Wash Buffer A (Biosearch Technologies, SMF-WA1-60), and then resuspended in 40 μl of 50% formamide Hybridization Buffer (Biosearch Technologies, SMF-HB1-10) containing 1.25 μM PP7 FISH probe. Hybridized samples were incubated at 30 °C for 16 h, and then stored at 4 °C for 6 weeks. Cells were washed 3 times with 50% formamide Wash Buffer A, resuspended in 100 μl DAPI at 10 μg/ml, and incubated at 30 °C for 30 min to label DNA. Cells were then washed in 500 μl Wash Buffer B (Biosearch Technologies, SMF-WB1-20), and resuspended in 5 μl freshly-filtered 2× SSC buffer (Ambion, AM9763) for imaging.

**FISH assay**. For each sample, 2 μl of cells was pipetted onto a #1 coverslip (45 mm × 50 mm, Fisher Scientific, #12-544 F). A 1.5% agarose pad was placed on top of the sample droplet to press the cells onto the imaging surface, and another, smaller, #1 coverslip (22 mm × 22 mm, Fisher Scientific, #12-545B) was placed on top of the agarose pad. Imaging was performed using an inverted epifluorescence microscope (Zeiss Axio Observer.Z1) with a 100× 1.46 NA oil-immersion phase-contrast objective lens (Zeiss, alpha Plan-Apochromat Ph3 M27) and a cooled digital CMOS camera (Hamamatsu Orca Flash 4.0). Zen Pro software was used to control microscope and camera. Five channels were collected in the following sequence, from longest to shortest wavelength to minimize effects of cross-talk. In Channel 1, TAMRA fluorescence was collected using excitation from an HXP 120 W mercury arc lamp at 100% intensity, with a 550 ± 12 nm excitation filter, a 570 nm beamsplitter, and a 605 ± 35 nm emission filter. For this channel, 9 z-slices were collected at a spacing of 200 nm per slice (total z-range of 1.6 μm), with an integration time of 1 s per slice. In Channel 2, YFP fluorescence was collected using LED excitation at 470 nm (Zeiss Colibri, 100% intensity), with a 470 ± 20 nm excitation filter, a 495 nm beamsplitter, and a 525 ± 25 emission filter, at a single z-slice with an integration time of 1 s. In Channel 3, CFP fluorescence was collected using excitation from an HXP 120 W mercury arc lamp at 100% intensity, with a 436 ± 12 nm excitation filter, a 455 nm beamsplitter, and a 480 ± 20 nm emission filter, at a single z-slice with an integration time of 1 s. In Channel 4, DAPI fluorescence was collected using LED excitation at 385 nm (Zeiss Colibri, 25% intensity), with a 359 ± 24 nm excitation filter, a 395 nm beamsplitter, and a 445 ± 25 emission filter, at a single z-slice with an integration time of 50 ms. In Channel 5, phase contrast was used to image bacterial cell bodies using a halogen lamp set to 4 V, collected over 9 s-slices separated by 200 nm each (total z-range of 1.6 μm), with an integration time of 100 ms per slice. Each sample was imaged at a minimum of 3 different locations. Images were exported as TIFF files for subsequent analysis.

**Measurement of cell growth**. Overnight cultures of strains were grown in the same manner as for the microscope assay. Briefly the overnight culture was diluted 1:178 into 200 μl M9 medium in a V-bottom 96-well plate, and grown at 37 °C at 1000 r.p.m. in an ELMI shaker for three hours. Then the cultures were diluted 1:667 into 1 ml M9 medium with antibiotics and 6 ng/ml aTc in a deep 96-well plate and grown at 37 °C in an INFORS-HT shaker. Starting from 4 h after incubation, the $OD_{600}$ of the sample was measured every 20 min in a plate reader (Synergy H1 microplate reader, Biotek) for 2 h. The doubling time was calculated by assuming exponential growth in this time period.

**Reporting summary**. Further information on research design is available in the Nature Research Reporting Summary linked to this article.

## Data availability

Source data are provided with this paper. Genetic part sequences are available in Supplementary Information. Plasmids are available from Addgene. Any other relevant data are available from the corresponding author upon reasonable request.

## Code availability

Matlab scripts used for image processing are released as open – source software under the MIT license (GitHub repository: https://github.com/VoigtLab/Promoter_Activity_Quantification).

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

## Acknowledgements

This work was supported by US National Institutes of Standards and Technology grant no. 70-NANB16H164 (B.S., J.R. and C.A.V.), US Department of Energy grant no. DE-FOA-0001650 (B.S. and C.A.V.), and the National Research Council Postdoctoral Associateship (J.R.). The National Institute of Standards and Technology notes that certain commercial equipment, instruments, and materials are identified in this paper to specify an experimental procedure as completely as possible. In no case does the identification of particular equipment or materials imply a recommendation or endorsement by NIST, nor does it imply that the materials, instruments, or equipment are necessarily the best available for the purpose.

## Author contributions

C.A.V., B.S. and D.R. conceived the study, designed the experiments and wrote the manuscript. B.S. and D.A.A. constructed the plasmids and carried out microscope experiments. J.R., B.S. and N.A. performed FISH experiments.

## Competing interests

The authors declare no conflict of interests.
