## [Peer Review File · Nature Communications]

Reviewers' Comments:

Reviewer #1:

Remarks to the Author:

In "Single-cell measurement of plasmid copy number and promoter activity," the authors present a method to quantify plasmids, transcripts, and protein levels in live cells (*E. coli*). Specifically, the authors use quantitative fluorescence microscopy to quantify plasmid copy number by using the PhiF system (RFP), mRNA transcripts by using the PP7 system (CFP), and proteins by using YFP fusion reporter. The main scientific contribution of this manuscript is to demonstrate the use of these three live-cell techniques to achieve a measurement of the strength of the gene promoter in terms of the flux of RNA polymerases initiation from the promoter per unit of time. This quantitative measure for the strength of the gene promoter is more precise than traditional measurements that characterize promoter strengths in arbitrary units.

The authors show that they are able to quantify the distributions of the plasmid copy number, using multiple plasmids containing different origins of replication, and genetic modifications are used as controls to ensure the results are general and not influenced by specific elements on the plasmids' sequences. The promoter activity was quantified in units of RNAP flux, using a function dependent on the promoter copy number, mRNA copy number, and RNA degradation rate. Promoter copy numbers and mRNA copy numbers were measured on each cell, and rifampicin assays are used to quantify the population-averaged transcript mean half-life. Additionally, promoter activity was quantified for different cell media, and cell volume.

A simple mathematical model was implemented to determine plasmid replication and replication mechanisms. A second 2-state promoter model was solved by the FSP method and was implemented to fit the distribution of transcripts as a function of the plasmid copy number. The authors show that these models can be fit well to the data.

Overall, the short article is well-structured and well-written, and it addresses an interesting and important topic in its field (i.e., the simultaneous imaging of the DNA->RNA->protein dynamics in single living cells). Obtained results including plasmid copy number, mRNA copy number, and promoter activity are in agreement with previously reported numbers and have been well-verified through appropriate controls using single-molecule fluorescence in situ hybridization (smFISH). The presented technology could have a substantial impact on the field, and could potentially allow for more comprehensive live-cell probing of the basic mechanisms of the central dogma in *E. coli*. However, the current manuscript does not seem to really push the technology past the existing capabilities of fixed-cell methods (e.g., single molecule FISH or single-cell immunofluorescence staining). Although we agree with the authors that such capabilities should be possible, it needs to be demonstrated explicitly. Without a clear demonstration that the proposed technology can provide new insight or more accurate measurements of live cell activities in real time, the impact this manuscript will be substantially less than that expected from a methods paper in *Nature Communications*. Detailed comments are given below.

Major comments:

1) The main advantage of quantitative fluorescence microscopy and live cell single-molecule measurements is to achieve coupled time course intensity trajectories for the studied species (DNA, transcripts, and/or proteins). Indeed, this is the main conclusion of the manuscript (Lines 171 and 172 – "In a single experiment, our approach is able to visualize the number of plasmids, mRNA, and protein expression levels in live cells"). Unfortunately, the present manuscript doesn't actually demonstrate any of the special advantages of live-cell quantitative microscopy techniques. All the presented measurements appear to be conducted at specific time points with fixed cells -- and no time courses are shown in this manuscript. The presented experiments could all be conducted equally well using single-molecule DNA or RNA FISH and single-cell immunofluorescence, which have been used extensively for single-cell and single-molecule

measurements at specific instances in time, and without genetic manipulation to add PP7, PhiF, or fluorescent proteins. Although it is much appreciated that the authors use FISH as a crucial control to verify their methods, they also need to show that their approach gives more insight than that achievable by fixed cell techniques. Specifically, the authors need to provide examples of time-lapse single-cell movies and single-cell time course analyses for plasmid, transcript, and protein intensities.

2) The authors' computational analyses should also be extended somewhat to take advantage of the opportunities provided by time-lapse, single-cell fluorescence microscopy experiments. For example, can RNAP flux be quantified more directly by using the temporal measurements of nascent transcription provided by live cell fluorescence imaging? Can cell-to-cell variability in this flux be quantified? Can the auto- and cross-correlations between plasmid, mRNA and protein content be quantified and compared in different genetic or environmental circumstances? Do plasmids or plasmid clusters switch between active states (with nascent mRNA) and inactive states (without), what are the distributions of switching times, and does that depend on plasmid cluster size? Can the authors quantify the asymmetry in the cross-covariance functions between plasmid numbers, mRNA numbers, and protein levels — one would expect that the correlations should be strongest at some non-zero lag time. Given the brevity of the paper, a full computational investigation of these temporal characteristics is not necessary and may be best left for a more thorough computational investigation, but if the authors could provide some basic examples of new types of information that can be measured and modeled using the authors' methods, it could help motivate future adoption of the technology and attract follow up investigations from other computational groups.

3) Line 113-115, "From this, the intensity corresponding to a single transcript can be calculated and used to normalize the data to calculate the absolute number of transcripts," is not very clear. Are the authors talking about total number of transcripts in a specific spot or in the entire cell? It seems that this is appropriate to determine the number per spot, which could then be summed to find the number per cell.

Minor comments:

3) Line 132: define 'RNAP/s-DNA'.

4) The authors should include a statement on the availability of the computational codes used in the manuscript.

5) Figure S4. Panel b. Use legends for the red and blue lines. Panel c. Use labels to denote the specific plasmids.

6) Figure S6. Panel a. Consider using a legend for the labels for the promoters.

Reviewer #2:

Remarks to the Author:

The manuscript entitled "single cell measurement of plasmid copy number and promoter activity" describes a new method to evaluate in *Escherichia coli* plasmid copy number at single cell level using a Fluorescent Reporter Operator System (FROS), combined with a PP7 based system to measure mRNAs numbers and fluorescent protein reporter for protein production. The combination of these three readouts allows the authors to infer the activity of a promoter in absolute unit as opposed to most common methods that rely on comparative results.

This is an important manuscript that addresses a question that has limited the Systems and Synthetic biology community for a long time. Indeed, despite recent progress in developing chromosomal landing pads to integrate synthetic circuits directly on the chromosome, plasmids are

still largely used as vectors to introduce synthetic circuits in bacteria and their copy number needs to be assessed rigorously.

The manuscript is very well written and presents a large amount of quantitative data that support the claims of the authors regarding their capacity to produce estimates of promoter activity for several plasmids. It relies on a combination of methods that have already been established (FROS, PP7 etc..) but the novelty lies in the carefully crafted use of these methods in single cells at the same time.

Here are my general comments regarding the manuscript:

-The manuscript is written in a very condensed manner which makes reading it difficult. Some results/datasets need to be described in a more detailed manner so that it is easier to understand why the experiment was performed and what information it brings (I give below several examples but they are not exhaustive).

-Some data-set should be re-analysed to provide more information. The most important ones are comparing GFP and FROS data (for plasmid copy number estimation) as well as comparing RNA FISH data and PP7 results. See below for detailed suggestions.

In light of the current pandemic and given the extensive experimental data already provided, I have deliberately refrained from suggesting more experiments (although I can think about quite a few really cool things to do with this system!).

I however point where the authors should indicate in their manuscript limitations of the current work (in my opinion they are anyway minor).

Major comments

-Please expand the introduction to clearly state why counting plasmids copy number in single cells is important and challenging (the case for the importance of promoter activity measurement is well made, although expanding on why absolute quantification if useful would be welcome). Please discuss also the importance in the context of more and more chromosomal landing pads systems (some of them actually developed by the Voigt lab).

- why use NEB10-beta? This is a cloning strain with multiple mutations in it (including recA and relA/spoT) which is not particularly suitable for physiological measurement in E. coli. I know the recA mutation is believed to stabilize some plasmids (although I am not certain this is actually true), is this a reason behind the choice? Do the authors have sense of whether the genetic background of the strain might affect the method/results?

-plasmid copy number quantification by FROS: please explain why it is expected to see spots that may actually be more than one plasmid and therefore spot intensity needs to be calculated. I know this has already been observed, but it would be useful for non plasmid-specialists to explain the reasoning step by step.

-the mathematical model needs more explanation in the main text. It is still unclear to me why this is converging to distribution where a significant number of cells do not carry a plasmid (1-5% experimentally), and a bit of text to explain the intuition would be useful. Moreover, the comparison of simulation and experimental data in supplementary figure 4c suggests that there are more cells with no plasmid detected experimentally than predicted by the model at least for the pUC plasmid; but I could not find any discussion of that point.

-Regarding this results (plasmid free cells) : I could not find information as to whether there are GFP expressing cells for which no PhIF-focus is detected (which would constitute a false negative). The correlation of spot intensity with GFP intensity (Sup Figure 2c) is good but it is not sufficient to rule out lack of plasmid detection in a few cells. What is needed is GFP intensity for all cells, including those that do not have a spot. This the first main analysis that I think should be added to the paper. To be clear: if this analysis suggests some plasmids are not detected, that is not a problem per se (no method is perfect) but this should be clearly stated.

-analysis of RNA production: it is very nice that the authors compared the results obtained by FISH with those obtained by PP7-CFP (Fig 1h). However, I could only see some overlay of images but no

quantification of the number of transcript identified by FISH compared to PP7-CFP. This is the second analysis that I would suggest should be added to the paper. On that note, cell fixation to perform FISH is likely to impact the CFP signal, so I would expect FISH to be more sensitive than PP7-CFP on this experimental set-up. This will somewhat limit the comparison, but I think it should still be useful.

Ideally, distribution based on FISH data or live PP7CFP-data could be compared directly for the same plasmids/promoter pairs but unless these datasets are already available, I think the comparison of PP7-CFP/FISH on the same fixed cells should be enough.

-line 150 discussion on the bimodality: could the authors comment on why bimodality is observed for mRNA and not protein? My intuition is that for a highly expressed stable fluorescent reporter, fluctuations on mRNAs (which have a much short lifetime) should be averaged out at the protein level. I think this could be more clearly stated which would also contribute to explain the importance of the work presented in the article.

-line166, I see how the 100 fold change in copy number can be reduced to 4 fold at the RNA level if RNAPS are limiting but the reduction to 2 fold in protein is less clear to me. Do the authors suggest ribosomes might be limiting too?.

-line167: indeed this results suggest that fluctuation would be buffered but also that effort to boost production by using high copy number plasmid are likely to ne be very successful. I think this should be emphasized more clearly as it is a really important results of the paper.

-line 171 the discussion of the limitations is super short and would benefit from a more detailed explanation of their implications; For example if the pseudo steady state approximation is not valid what are the implications on the results?

-Line 179. The data shown in supp figure 14 are very interesting and merit a much more detailed presentation . Being able to infer the "transcriptional burden" in number of RNAPs for a complex circuit is new and I would appreciate a more complete description of the data in the main text.

Minor comments:

-NEB10 β is sometimes referred to as NEB10B (e.g. line 205)

-Figure 1f (description of the mathematical model): I think the red crosses in the first line indicate cells that are randomly removed from the pool. If so, make the crosses bigger so that they cover the cells entirely to make it easier to understand.

-line 120: I am not sure why the sgRNA detection is stated there? I would suggest possibly adding to the end of the discussion to describe further exciting work that can be done with this technique.

-I thank the authors for clearly stating in line 137 that they are using an approximation for mRNA degradation rate.

-figure 2b :how many cells were analyzed?

-the GFP signal in Supp figure 2b (top row, colE1 plasmid) is quite heterogeneous compared to figure 1b. Could the authors comment on that?

-I do not understand the fit in suppl. Figure 5c. Was the fit done with only two points (the blue ones)? In this case its RBS P2 and not P1? Please clarify.

-supp figure 6b: the YFP signal in the top row is very high which probably leads to quantification errors at single cell level because of point spread function error. I don't think this should impact the average reported in panel suppl. fig. 6a, but the calculation should be more clearly described and the caveat stated.

-supp figure 9: was the mRNA after rifampicin addition estimated also from the FISH data? please add it if possible for comparison purposes;

-Supp figure 11 typos in line 2 and 4 of the legend ("tripartate" and "fluorecence")

-supp figure 14a typo in panel a "senors" instead of sensor.

Dear Editor,

Please find above my review of the paper entitled "single cell measurement of plasmid copy

number and promoter activity". As mentioned in my remarks to the authors, I have purposefully refrained from suggesting more experiments in light of the current crisis. Please note that I did not take this decision as a favor to the authors (whom I do not know at all) but because I weighed carefully how much more information could be gained by additional experiments compared to the burden of performing them at the moment. Also the authors have included a lot of very useful information in the supplementary material some of which should be better presented and discussed to support their claims.

Please do not hesitate to contact me if you need further clarification.

Reviewer #1:

1. *The main advantage of quantitative fluorescence microscopy and live cell single-molecule measurements is to achieve coupled time course intensity trajectories for the studied species (DNA, transcripts, and/or proteins). Indeed, this is the main conclusion of the manuscript (Lines 171 and 172 – “In a single experiment, our approach is able to visualize the number of plasmids, mRNA, and protein expression levels in live cells”). Unfortunately, the present manuscript doesn’t actually demonstrate any of the special advantages of live-cell quantitative microscopy techniques. All the presented measurements appear to be conducted at specific time points with fixed cells -- and no time courses are shown in this manuscript. The presented experiments could all be conducted equally well using single-molecule DNA or RNA FISH and single-cell immunofluorescence, which have been used extensively for single-cell and single-molecule measurements at specific instances in time, and without genetic manipulation to add PP7, PhiF, or fluorescent proteins. Although it is much appreciated that the authors use FISH as a crucial control to verify their methods, they also need to show that their approach gives more insight than that achievable by fixed cell techniques. Specifically, the authors need to provide examples of time-lapse single-cell movies and single-cell time course analyses for plasmid, transcript, and protein intensities.*

New experiments have been performed to obtain time-lapse single cell measurements (new Figure S12). The text has also been edited. However, we disagree that movies are the primary value of this technique and it was not the motivation behind this study. The techniques described above cannot be performed simultaneously in a single cell and/or require fixing the cell. (Note that our fluorescence experiments are not performed with fixed cells.) Because of this, we can calculate the distribution of plasmid copy numbers across living cells and use the measured values to calculate the distribution of promoter activities across living cells. Neither of these measurements have been previously reported.

2. *The authors’ computational analyses should also be extended somewhat to take advantage of the opportunities provided by time-lapse, single-cell fluorescence microscopy experiments.*

Most of requested analyses in this comment we either cannot do because of technical limitations or it would take the paper in a different direction. Note that we are engineers and thus use these data in a way that is a little different than a physicist. We have already made use of the key numbers from this paper for simulations of genetic circuit design, which allows us to convert between arbitrary units to absolute units, for simulations of genetic circuit design. This appears in the paper, “Genetic circuit characterization by inferring RNA polymerase movement and ribosome usage,” Nature Communications 11, 1-18, which was submitted about the same time and has been published (Figure 5 of that paper). We can also use these numbers to be quantitative about resource utilization, which we outline in the current paper being discussed.

2A. *For example, can RNAP flux be quantified more directly by using the temporal measurements of nascent transcription provided by live cell fluorescence imaging? Can cell-to-cell variability in this flux be quantified?*

No, we are not able to track nascent transcription. We do calculate the cell-to-cell variability of the flux from the promoter and this is presented as a distribution (but it is not nascent).

2B. Can the auto- and cross-correlations between plasmid, mRNA and protein content be quantified and compared in different genetic or environmental circumstances?

We have calculated the correlations between DNA, mRNA and protein, which are shown in Figure 2b. New data have been added for our system across environments (media types) and genetic backgrounds (new Figure 3). Note that these measurements are not from time lapse movies and we were not able to obtain the auto- and cross-correlations, as envisioned in this comment.

2C. Do plasmids or plasmid clusters switch between active states (with nascent mRNA) and inactive states (without), what are the distributions of switching times, and does that depend on plasmid cluster size?

We are not able to calculate the switching between active and inactive states of plasmids.

2D. Can the authors quantify the asymmetry in the cross-covariance functions between plasmid numbers, mRNA numbers, and protein levels — one would expect that the correlations should be strongest at some non-zero lag time.

We are not able to calculate the cross-variance.

2E. Given the brevity of the paper, a full computational investigation of these temporal characteristics is not necessary and may be best left for a more thorough computational investigation, but if the authors could provide some basic examples of new types of information that can be measured and modeled using the authors' methods, it could help motivate future adoption of the technology and attract follow up investigations from other computational groups.

We do include a simple computational model to aid our understanding of the distributions in Figure 1 and Figure S4. We have edited the paper to better explain the mathematical underpinnings of this model (Supplementary Note 1).

3. Line 113-115, "From this, the intensity corresponding to a single transcript can be calculated and used to normalize the data to calculate the absolute number of transcripts," is not very clear. Are the authors talking about total number of transcripts in a specific spot or in the entire cell? It seems that this is appropriate to determine the number per spot, which could then be summed to find the number per cell.

We are referring to the total number of transcripts in a specific spot. We have modified the text for clarity.

4. Line 132: define 'RNAP/s-DNA'.

The text has been edited to include the definition of “RNAP/s – DNA”, which is “RNA polymerases per second per DNA.” It is the flux of RNAP on a DNA strand.

5. The authors should include a statement on the availability of the computational codes used in the manuscript.

We have included a code availability statement in the manuscript. Per lab policy, all code we produce is open source and provided on Github.

6. Figure S4. Panel b. Use legends for the red and blue lines. Panel c. Use labels to denote the specific plasmids.

We have added legends to Figure S4. Plasmid labels are also added in panel c.

7. Figure S6. Panel a. Consider using a legend for the labels for the promoters.

We have added a legend in panel a of Figure S6.

Reviewer #2:

1. Please expand the introduction to clearly state why counting plasmids copy number in single cells is important and challenging (the case for the importance of promoter activity measurement is well made, although expanding on why absolute quantification if useful would be welcome). Please discuss also the importance in the context of more and more chromosomal landing pads systems (some of them actually developed by the Voigt lab).

The introduction has been edited as suggested.

2. Why use NEB10-beta? This is a cloning strain with multiple mutations in it (including *recA* and *relA/spoT*) which is not particularly suitable for physiological measurement in *E. coli*. I know the *recA* mutation is believed to stabilize some plasmids (although I am not certain this is actually true), is this a reason behind the choice? Do the authors have sense of whether the genetic background of the strain might affect the method/results?

New experiments have been performed to show measurement in the “more native” *E. coli* strain MG1655 and the fast-growing marine bacterium *Vibrio natriegens*. The results are shown in Figure S15 and the main text (new Figure 3). Indeed, the plasmid loss is higher when *recA* and *relA/spoT* are present.

3. Plasmid copy number quantification by FROS: please explain why it is expected to see spots that may actually be more than one plasmid and therefore spot intensity needs to be calculated. I know this has already been observed, but it would be useful for non plasmid-specialists to explain the reasoning step by step.

We have added discussion in the text.

4. The mathematical model needs more explanation in the main text. It is still unclear to me why this is converging to distribution where a significant number of cells do not carry a plasmid (1-5% experimentally), and a bit of text to explain the intuition would be useful. Moreover, the comparison of simulation and experimental data in supplementary figure 4c suggests that there are more cells with no plasmid detected experimentally than predicted by the model at least for the *pUC* plasmid; but I could not find any discussion of that point.

The mathematical model converges to a steady state distribution that is shaped by the tightness of plasmid replication control and plasmid partitioning due to cell division. Plasmid loss occurs when the number of plasmids is small and partitioning is less evenly than binomial prediction, thus leaving one daughter without plasmids. The text has been edited to better explain the difference between predicted and experimental value. We hypothesize it is due to plasmid clustering effects or active partition mechanisms that may not be captured by a single parameter in our model.

5. Regarding this result (plasmid free cells): I could not find information as to whether there are GFP expressing cells for which no PhIF-focus is detected (which would constitute a false negative).

The correlation of spot intensity with GFP intensity (Sup Figure 2c) is good but it is not sufficient to rule out lack of plasmid detection in a few cells. What is needed is GFP intensity for all cells, including those that do not have a spot. This the first main analysis that I think should be added to the paper. To be clear: if this analysis suggests some plasmids are not detected, that is not a problem per se (no method is perfect) but this should be clearly stated.

The suggested analysis has been added to Figure S2 (f-i). We find that in the cells with low PhIF-RFP spot intensity, the GFP fluorescence is also very low. This suggests that the dominant contribution to the reported value is plasmid loss.

6. Analysis of RNA production: it is very nice that the authors compared the results obtained by FISH with those obtained by PP7-CFP (Fig 1h). However, I could only see some overlay of images but no quantification of the number of transcript identified by FISH compared to PP7-CFP. This is the second analysis that I would suggest should be added to the paper. On that note, cell fixation to perform FISH is likely to impact the CFP signal, so I would expect FISH to be more sensitive than PP7-CFP on this experimental set-up. This will somewhat limit the comparison, but I think it should still be useful.

Ideally, distribution based on FISH data or live PP7CFP-data could be compared directly for the same plasmids/promoter pairs but unless these datasets are already available, I think the comparison of PP7-CFP/FISH on the same fixed cells should be enough.

The correlation of the signal (number of transcripts) measured using PP7-CFP versus FISH for the same fixed cells is shown in Figure S6g. The correlation between the two measurements is strong ($R^2 = 0.95$) and the FISH sensitivity is only slightly higher.

7. Line 150 discussion on the bimodality: could the authors comment on why bimodality is observed for mRNA and not protein? My intuition is that for a highly expressed stable fluorescent reporter, fluctuations on mRNAs (which have a much short lifetime) should be averaged out at the protein level. I think this could be more clearly stated which would also contribute to explain the importance of the work presented in the article.

Indeed, this is our interpretation as well and we have edited the text to clarify.

8. Line166, I see how the 100-fold change in copy number can be reduced to 4-fold at the RNA level if RNAPs are limiting but the reduction to 2-fold in protein is less clear to me. Do the authors suggest ribosomes might be limiting too?

Yes, this could be due to the additional resource limitation imposed by a fixed number of ribosomes per cell, especially noting that the number mRNA transcripts from a plasmid will be relatively high.

9. Line167: indeed this results suggest that fluctuation would be buffered but also that effort to boost production by using high copy number plasmid are likely to not be very successful. I think this should be emphasized more clearly as it is a really important results of the paper.

We have edited the text to make this point.

10. *Line 171 the discussion of the limitations is super short and would benefit from a more detailed explanation of their implications; For example if the pseudo steady state approximation is not valid what are the implications on the results?*

Because mRNA and protein production and degradation are faster than plasmid replication and removal by cell division, the pseudo-state-state approximation holds for the analysis. The time it is most likely to be inaccurate is immediately after cell division, where the plasmid number per cell changes sharply and it requires time for the mRNA and protein levels to adjust. We have performed new experiments (time lapse movies) and have used these data to calculate the DNA and mRNA copy number as well as the promoter activity after cell division (Figure S12). Indeed, there is an initial increase in the promoter activity that then returns to steady-state. We have added these data and discussion to the text.

11. *Line 179. The data shown in supp figure 14 are very interesting and merit a much more detailed presentation. Being able to infer the “transcriptional burden” in number of RNAPs for a complex circuit is new and I would appreciate a more complete description of the data in the main text.*

We have expanded the main text to include more discussion of this point. Note that these data were used in the manuscript, “Genetic circuit characterization by inferring RNA polymerase movement and ribosome usage” (Nature Communications, 2020, 11:1), to present circuit RNAP fluxes and cell burden in absolute units.

12. *NEB10-beta is sometimes referred to as NEB10B (e.g. line 205)*

This has been corrected.

13. *Figure 1f (description of the mathematical model): I think the red crosses in the first line indicate cells that are randomly removed from the pool. If so, make the crosses bigger so that they cover the cells entirely to make it easier to understand.*

We have changed Figure 1f to make the crosses bigger.

14. *Line 120: I am not sure why the sgRNA detection is stated there? I would suggest possibly adding to the end of the discussion to describe further exciting work that can be done with this technique.*

We initially tried moving it to the end, but it did not integrate well, so we left it where it is.

15. *Figure 2b: how many cells were analyzed?*

We have edited the figure legend to include the numbers of cells analyzed.

16. *The GFP signal in Supp figure 2b (top row, colE1 plasmid) is quite heterogeneous compared to figure 1b. Could the authors comment on that?*

The pUC plasmid (top row, Figure S2b) produces more heterogeneous GFP expression as compared to the ColE1 plasmid in Figure 1b. The simulations predict that the pUC backbone has a looser copy number control mechanism compared to the ColE1 backbone (Table S1, parameter *K*). This would lead to a higher variation in copy number. We have edited the Supplementary Note 1 to more clearly describe the math underlying the simulations.

17. *I do not understand the fit in suppl. Figure 5c. Was the fit done with only two points (the blue ones)? In this case its RBS P2 and not P1? Please clarify.*

The linear fit in Figure S5c was performed with three points: (0, 0) and the two data points from RBS P2. The caption has been edited for clarity.

18. *Supp figure 6b: the YFP signal in the top row is very high, which probably leads to quantification errors at single cell level because of point spread function error. I don't think this should impact the average reported in panel suppl. fig. 6a, but the calculation should be more clearly described and the caveat stated.*

The caption has been edited for clarity, including this point.

19. *Supp figure 9: was the mRNA after rifampicin addition estimated also from the FISH data? please add it if possible for comparison purposes;*

The mRNA decay rate was estimated from the decay of PP7-CFP spot intensity and it is consistent with previous mRNA decay measurements. Quantifying the mRNA number from the FISH signal in this work would require a large amount of additional experimentation.

20. *Supp figure 11 typos in line 2 and 4 of the legend ("tripartate" and "fluorecence")*

This has been corrected.

21. *Supp figure 14a typo in panel a "senors" instead of sensor.*

This has been corrected.

Reviewers' Comments:

Reviewer #1:

Remarks to the Author:

We were satisfied with the revised manuscript and particularly appreciate the addition of the time lapse experiments, which clearly demonstrate a crucial improvement of the proposed technology over previous fixed-cell capabilities (i.e, capability to simultaneously measure plasmids, mRNA and proteins in living cells). This technology should be highly valuable for the quantitative characterization of dynamics for different synthetic gene regulatory modules. The modified text in the introduction, results, and supplementary sections are appropriate in that they better explain the importance of the study; results were extended to include new experiments, and SI was expanded to describe with more detail the mathematical model. A lot remains to be done in terms of rigorous integration of computational models with these multiplexed single-cell quantitative data, but that is to be expected for any new experimental technique. (We count this as a strong positive aspect of the current manuscript, and we look forward to seeing future progress in that direction.) We have no further major or minor comments.

Reviewer #2:

Remarks to the Author:

The revised version of the manuscript entitled "Single-cell measurement of plasmid copy number and promoter activity" addresses all my previous comments. I particularly appreciate the Authors' effort to add more data and to perform the additional analyses that I requested. Indeed the correlation between the PP7-CFP signal and FISH data to measure mRNAs is very strong and convincing. Similarly, the comparison of GFP signal and PLF-RFP spots convincingly suggests that most if not all plasmids are detected.

The new experimental data, in particular the time-lapse experiments are very interesting and show convincingly how promoter activity can be measured in real time.

I also find the new version of the manuscript with the expended introduction and discussion much easier to read.

I have noted the following typos/minor comments (lines numbers refer to the new version of the manuscript):

-line 137: "bimodality has been observed for E. coli promoters". Add other -> "bimodality has been observed for OTHER E. coli promoters".

-Line 179 "the plasmid and mRNA copy number increase after cell division, as expected". I think the authors mean "decrease as expected".

-Line 204 "it speaks to the futility of using high copy number plasmids increase heterologous protein expression" a "to" is missing. "it speaks to the futility of using high copy number plasmids TO increase heterologous protein expression".

-legend supp figure 12: typo "Cells were tranfromated with pSB223" -> "Cells were transformed"

legend supp figure 13: "These data correspond to that show in Figure 2a,"-> "These data correspond to that showN in Figure 2a"

Reviewer #2:

1. Line 137: *"bimodality has been observed for E. coli promoters". Add other -> "bimodality has been observed for OTHER E. coli promoters".*

This has been corrected.

2. Line 179 *"the plasmid and mRNA copy number increase after cell division, as expected". I think the authors mean "decrease as expected".*

This has been corrected.

3. Line 204 *"it speaks to the futility of using high copy number plasmids increase heterologous protein expression" a "to" is missing. "it speaks to the futility of using high copy number plasmids TO increase heterologous protein expression".*

This has been corrected.

4. Legend supp figure 12: typo *"Cells were tranfromated with pSB223" -> "Cells were transformed"*

This has been corrected.

5. Legend supp figure 13: *"These data correspond to that show in Figure 2a,"-> "These data correspond to that showN in Figure 2a"*

This has been corrected.